# Improved Performance of Asphalt Concretes using Bottom Ash as an Alternative Aggregate

**Apinun Buritatum** [1,2,3], **Apichat Suddeepong** [1,2,3], **Suksun Horpibulsuk** [1,2,3,4,5,*], **Kongsak Akkharawongwhatthana** [1,3], **Teerasak Yaowarat** [1,2,3], **Menglim Hoy** [1,2,3,4], **Chalermphol Bunsong** [6] and **Arul Arulrajah** [7]

1. Center of Excellence in Innovation for Sustainable Infrastructure Development, Suranaree University of Technology, Nakhon Ratchasima 30000, Thailand; apinun_ce@hotmail.com (A.B.); suddeepong@g.sut.ac.th (A.S.); sam.kongsak@gmail.com (K.A.); teerasakyaowarat@gmail.com (T.Y.); menglim@g.sut.ac.th (M.H.)
2. School of Civil and Infrastructure Engineering, Suranaree University of Technology, Nakhon Ratchasima 30000, Thailand
3. Graduate Program in Civil Engineering and Construction Management, Suranaree University of Technology, Nakhon Ratchasima 30000, Thailand
4. School of Civil Engineering, Suranaree University of Technology, Nakhon Ratchasima 30000, Thailand
5. Academy of Science, The Royal Society of Thailand, Bangkok 10300, Thailand
6. Product Development Section, Coal Combustion Product Business Department, Business Management Division, Electricity Generating Authority of Thailand (EGAT), Nonthaburi 11130, Thailand; chalermphol.b@egat.co.th
7. Department of Civil and Construction Engineering, Swinburne University of Technology, Melbourne, VIC 3122, Australia; aarulrajah@swin.edu.au
* Correspondence: suksun@g.sut.ac.th

**Abstract:** Road networks are major infrastructures that support the economic development in both developed and developing countries. Bottom ash (BA) is a by-product from coal-fired powerplants, which is composed of a lipophilic molecule with effective reactivity to bituminous binder. BA was adopted in this research, as a green fine aggregate, to improve the mechanistic performance of asphalt concretes in this research. The effect of BA-replacement ratio (0%, 10%, 15%, 20% and 25%, by total weight of natural fine aggregate) on the Marshall stability and flow, indirect tensile strength (ITS), and mechanistic performance of BA-asphalt concrete, were examined. The mechanistic performance tests included the indirect-tensile condition (indirect tensile resilient modulus (IT $M_r$), indirect tensile fatigue life (ITFL)) and compressive condition (permanent deformation (PD), rut depth, and skid resistance). BA replacement improves the Marshall stability and flow, strength index, and ITS, up to the optimum BA-replacement ratio, of 5%. The change in IT $M_r$ was found to be linearly proportional to the change in ITS, for all BA-replacement ratios. The ITFL is dependent upon the repeated stress level and can be estimated in terms of IT $M_r$. For the compressive condition, the PD, rutting, and skid resistances were found to be improved by the BA replacement. The lowest PD and rut depth as well as the highest skid resistance, for IT $M_r$ and ITFL, were found at the optimum BA-replacement ratio, of 5%. The outcome of this research will promote the usage of BA as a cleaner additive in asphalt concrete pavement, which is useful in terms of engineering and environmental perspectives.

**Keywords:** recycled material; bottom ash; mechanistic performance; asphalt concrete

## 1. Introduction

Electricity power generation in many countries worldwide relies on the process of fuel combustion. Fossil fuels, such as petroleum and coal, are used as the primary resource of conventional power generation. Alternative-green-energy generation technologies, using biomass, solar, and wind power, can successfully substitute the utilization of fossil fuels for energy generation. However, due to the current economic cost advantage of fossil fuels, such as coal, they remain a major source of energy for the electricity generation in many countries, including Thailand. The pollution due to coal-fired power plants, mostly, comprises noxious fumes and dust, which are emitted during the combustion process. This

emission contributes significantly to global warming and causes long-term health problems, for people in the vicinity of the power plant [1–4].

In order to reduce pollution, the residues from coal combustion are treated with a pollution-collector system, such as precipitators or baghouses, before the release of treated air into the atmosphere. After the pollution-treatment process, the residue by-products are turned into Coal Combustion Products (CCP) including gypsum, fly ash (FA), and bottom ash (BA). Most power plants in developed and developing countries deal with CCPs, by means of disposal to landfills and ash ponds. A larger number of landfills and the disposal of CCP are required, due to a higher demand for power-generation capacity, therefore, the reclamation and management of disposal sites becomes a problem, in terms of economic and environmental perspectives. Moreover, the contamination of toxic leachate from the CCP disposal site causes a serious environmental problem in the vicinity areas [1,2,5–9].

Several researchers have attempted to maximize the usage of CCP to mitigate pollution, due to conventional waste disposal, and create its value [1,4,5,10]. FA is the most utilized CCP product in the industrial and engineering projects worldwide, due to its excellent pozzolanic reactivity, physical characteristics, and commercial value. In contrast, the utilization of BA is in its infancy, due to a lack of knowledge about and research on its usage in infrastructure works. The current practice of disposal of BA to landfills, furthermore, results in various environmental issues [1,4,9].

The quantity of BA, from the combustion process of the coal-fired power plants in Thailand, exceeds more than one million tons per year. The Electricity Generating Authority of Thailand (EGAT) allocates an annual budget of USD 180,000, to dispose its BA waste in the landfill. EGAT attempts to develop the best solution, to maximize BA utilization in road construction, in support of the country development policy of the Thailand government.

BA has been characterized as a porous aggregate, with low bulk density, rough particle shape, and high water absorption [1]. For the geotechnical properties, its California Bearing Ratio (CBR), hydraulic conductivity, shear strength, and chemical soundness of BA are equivalent to or better than those of natural aggregates [1]. BA is, therefore, traditionally used as a substitute for clayey soil, to develop a lightweight fill material. BA can improve swelling and shrinkage, due to the moisture change and shear strength of the substituted clayey soil. In addition, BA can improve the drainage capacity and reduce the consolidation time of the substituted clayey soil [4,6,7,11–17].

Previous research, also, reported that BA utilization is suitable for concrete and asphalt applications. The heavy metals in BA can be encapsulated in the concrete and asphalt matrixes; hence, the negative environmental impacts are significantly lower, when compared with geotechnical applications. The primary chemical compositions of BA are similar to those of fly ash, consisting of silica ($SiO_2$), alumina ($Al_2O_3$), and ferric oxide ($Fe_2O_3$), more than 80% by total components. BA is, mostly, applied to substitute fine aggregates in concrete, with the purpose of mechanical-strength improvement, due to its pozzolanic reactivity. At the early stage, concrete with substituted BA exhibits a lower strength than conventional concrete. Due to its pozzolanic reactivity, the long-term mechanical strength of concrete with substituted BA is, however, improved [5,6,9,10,18–24].

In 1976, it was reported that BA can be used as an alternative aggregate, to substitute for fine aggregate, in asphalt mixes [25]. BA can replace the fine natural aggregate, without any additional admixture. Due to its high porosity, BA aggregate requires a large amount of bituminous binder. BA particles compose a lipophilic molecule, with effective reactivity to bituminous binder. The reaction between BA particles and bituminous binder, consequently, results in thicker asphalt film, hence improving the cohesion strength between aggregate particles. As such, the improved cohesion strength contributes to superior engineering properties, compared to conventional asphalt concrete. The addition of BA, at the optimum content, can significantly enhance the Marshall stability and indirect tensile strength of the hot-mix asphalt concrete. Thereafter, the stability of hot-mix asphalt concrete decreases, with excessive BA content beyond the optimum content. The hot-mix asphalt, containing BA as the fine aggregate, has a higher potential to resist the fatigue damage benchmarked

with the conventional asphalt concrete. In addition, the resistance to moisture susceptibility of hot-mix asphalt concrete can be enhanced by the addition of BA. In field-construction works, asphalt concrete containing BA requires lower compaction energy than conventional asphalt concrete. In terms of environmental impact, the addition of BA reduces the toxic leachate from the asphalt concrete, significantly [26–31].

BA replacement, effectively, functions as a green additive for warm-mix asphalt concrete. BA has the potential to improve the mechanical strength of warm-mix asphalt. The addition of BA reduces the carbon monoxide emission of warm-mix asphalt concrete, up to 75%, compared to conventional warm-mix asphalt concrete [32]. In addition to coal-fired BA, municipal solid-waste-incinerator bottom ash (MSWI-BA), can, also, be used as pavement materials. The MSWI-BA has similar physical properties to the coal-fired BA, such as low densities and high-absorption properties. The MSWI-BA can be used as the intact material, for the partial replacement of fine aggregate, to improve the performance of hot-mix-asphalt-concrete application. Apart from the engineering properties, MSWI-BA can, effectively, reduce the release of alkaline and heavy metals in the hot mix of asphalt concrete [33–36].

Based on the mechanistic-design approach, for flexible pavement, the performance of asphalt concretes is evaluated under dynamic loads, to simulate traffic loading [36]. The resilient modulus, under repeated stress, controls the elastic properties of asphalt concretes and is applied as the primary design parameter, for pavement design. The repeated stress, due to traffic loading, typically generates plastic/irrecoverable deformation. The accumulated irrecoverable strain, ultimately, leads to pavement distress, such as fatigue cracking and rutting failure [37]. Therefore, the resilient modulus, fatigue life, and rutting resistance are prime factors assessing the performance of asphalt concrete [38–42].

According to the transport infrastructure report of 2018 [43], Thailand's road transportation system had a total length of 467,013 km, whereby more than 80% is flexible pavement. Therefore, the utilization of BA, to improve the performance of flexible pavements, is considered as a sustainable innovation in Thailand. However, the understanding of the performance of BA-modified asphalt concrete is vital for real-world applications.

Several researchers have reported on the advantages of utilizing synthetic polymers and natural fibers in modified asphalt concrete, via mechanistic performance tests [44–48]. However, the evaluation of BA performance in modified asphalt concrete, under complete performance tests, is, still, limited. This research, therefore, aims to study the influences of BA content on the performance of asphalt concretes under completed compressive and tensile stress, under both static and dynamic loading, and, also, to suggest the optimum BA content, which provides the best performance. The performance of BA-modified asphalt concrete was evaluated, via the indirect tensile resilient modulus (IT $M_r$), indirect tensile resilient fatigue, dynamic creep, rutting resistance, and skid resistance tests. Its engineering properties and performance were compared with those of conventional asphalt concrete. The outcome of this research will promote the utilization of BA as a cleaner pavement material, which is a sustainable solution for EGAT and the coal-fired power plants worldwide.

## 2. Materials

### 2.1. Bottom Ash (BA)

BA samples were supplied by EGAT. The basic properties and particle-size distribution of BA are summarized in Table 1 and Figure 1, respectively. The particle-size distribution and soundness of BA did not meet the standard to be used as fine aggregate in asphalt-wearing course, in accordance with the Thailand Department of Highways standard [48]. BA contained a high content of fine particles; particles smaller than 4.75 mm are greater than 50%. In other words, BA alone could not be used as fine aggregate in asphalt concrete. As such, it is used to substitute the natural fine aggregate in asphalt concrete. Figure 2 presents an image of moist BA. It is noted that BA particles were coagulated to clusters,

indicating that they react with water. In other words, BA is characterized as a hydrophobic material, which can have an excellent reaction with bitumen binder (lipophilic) [27].

**Table 1.** Basic engineering properties of fine aggregate.

| Properties | Limestone | BA | DH-S 408/1989 Specifications |
|---|---|---|---|
| Soundness (%) | 4.65 | 24.3 | ≤9% |
| Sand equivalent (%) | 73.6 | 53.1 | ≥50% |
| Specific gravity | 2.64 | 2.53 | - |

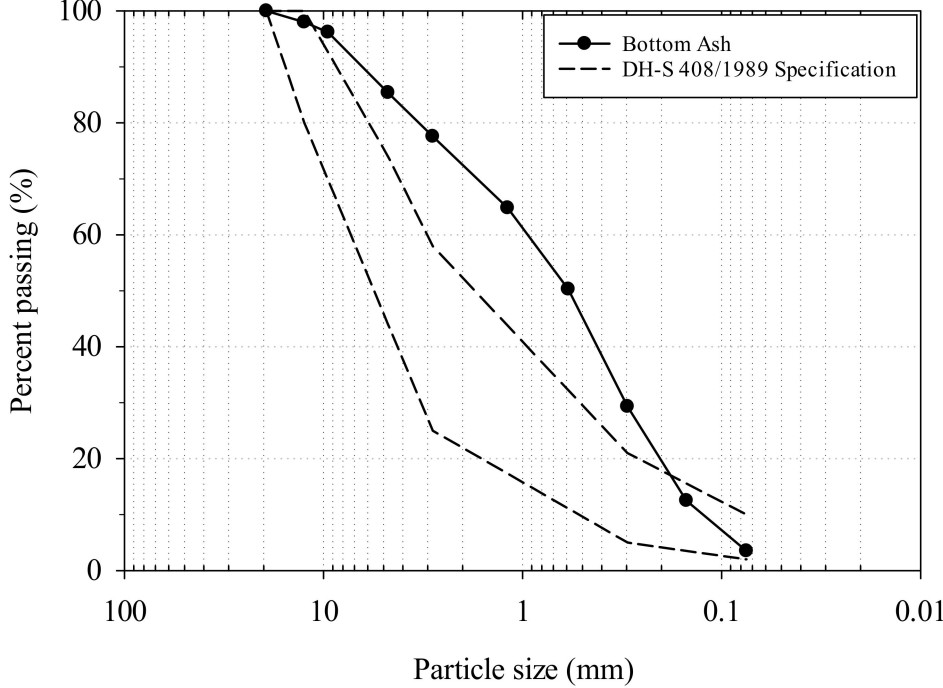

**Figure 1.** Particle size distribution of original BA aggregate.

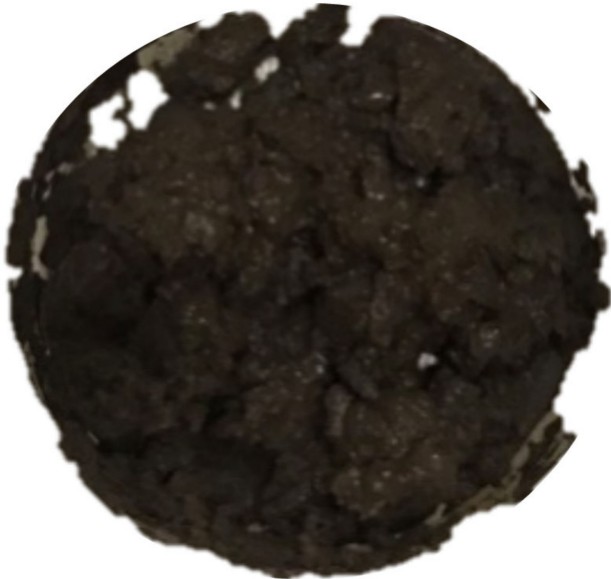

**Figure 2.** Image of moist BA.

## 2.2. Natural Aggregate

The studied natural aggregate was limestone, which is commonly used for conventional asphalt concretes in Thailand. The basic and engineering properties of fine and coarse aggregates are summarized in Tables 1 and 2, respectively. Based on the specification for pavement surface material selection [49], the properties of the studied limestone can be used for wearing surface course.

**Table 2.** Basic and engineering properties of coarse aggregate.

| Properties | | Limestone | DH-S 408/1989 Specifications |
|---|---|---|---|
| Soundness (%) | | 1.96 | 9% max |
| Los Angeles abrasion value, LA (%) | | 28.7 | 40% max |
| Elongation index (%) | Bin 2 | 20.5 | 30% max |
| | Bin 3 | 22.0 | |
| | Bin 4 | 24.9 | |
| Flakiness index (%) | Bin 2 | 19.9 | 30% max |
| | Bin 3 | 22.4 | |
| | Bin 4 | 23.6 | |
| Specific gravity | Bin 2 | 2.69 | - |
| | Bin 3 | 2.703 | - |
| | Bin 4 | 2.709 | - |
| Aggregate impact value, AIV (%) | | 21.3 | 25% max |
| Aggregate crushing value, ACV (%) | | 20.8 | 25% max |
| Asphalt absorption (%) | | 0.24 | - |

## 2.3. Bituminous Binder

The asphalt cement penetration grade, AC 60/70, commonly used for flexible pavement construction in Thailand, was selected as binder, to prepare both conventional and BA-asphalt concretes. Based on the standards for bituminous binder selection of the Thailand Department of Highways [50], the properties of the studied bituminous binder meet the minimum requirement, as demonstrated in Table 3.

**Table 3.** Basic and engineering properties of bituminous binder.

| Properties | Units | Specifications | Results |
|---|---|---|---|
| Penetration | - | 60–70 | 67 |
| Softening point | °C | 45–55 | 47.8 |
| Flash point | °C | >232 | 332 |
| Ductility at 25 °C | cm | >100 | 150 |
| Solubility in trichloroethylene | %wt | >99.0 | 99.97 |
| Elastic recovery | % | - | 35 |
| Specific gravity at 25 °C | - | - | 1.02 |
| Test on residue from thin-film oven test (5 h at 163 °C) | | | |
| Weight loss | % by wt. | <0.8 | 0.12 |
| Penetration | % by wt. | >54 | 71.1 |
| Ductility at 25 °C | cm | >50 | 150 |

## 3. Experimental Programs

### 3.1. Sample Preparation

The gradation of limestone aggregate was adjusted, to meet the boundaries specified by DH-S408/1989 (Figure 3) [49]. BA was passed through a 4.75 mm sieve, to remove coarser aggregate. The mass of fine limestone aggregate was replaced, by the mass of fine BA, at 0%, 10%, 15%, 20%, and 25% replacement ratios. Table 4 shows the mix ingredient at the studied replacement ratios. The properties tests of the combined aggregates, sand equivalent and soundness, at each BA-replacement ratio, were carried out, and the test results are summarized in Figure 4a,b. The solid line represents the specification requirement of DH-S408/1989. A higher BA-replacement ratio results in lower soundness and sand-equivalent value. It was evident that the soundness and sand equivalence of combined aggregate met the minimum requirement of DH-S408-1989 [49], for all BA-replacement ratios tested.

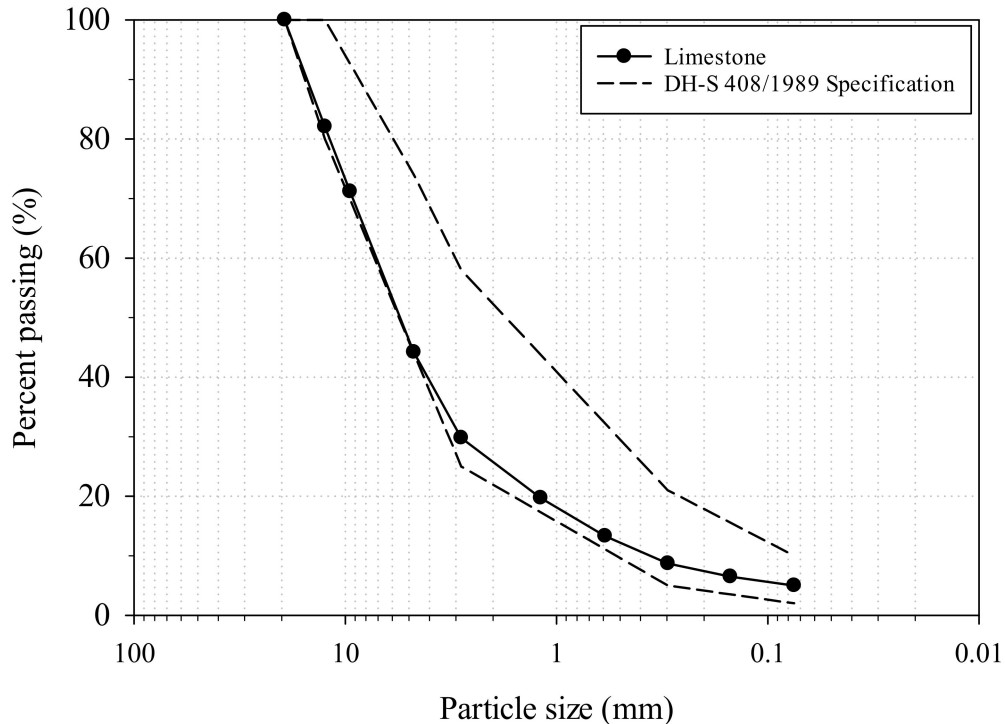

**Figure 3.** Particle-size distribution of limestone aggregate.

**Table 4.** Mix design of BA-asphalt concretes.

| Total Weight of Aggregate per Sample (g) | BA Replacement | | Limestone Aggregate | | | | | | |
| | | | Bin 1 | | Bin 2 | | Bin 3 | | Bin 4 | |
| | | | (4.75–0.075 mm) | | (4.75–0.6 mm) | | (12.5–2.36 mm) | | (19.0–4.75 mm) | |
| | % wt. of Bin 1 | Weight (g) | % wt. | Weight (g) | % wt. | Weight (g) | % wt. | Weight (g) | % wt. | Weight (g) |
|---|---|---|---|---|---|---|---|---|---|---|
| 1200.00 | 0 | 0.00 | 35.00 | 420.00 | 22.00 | 264.00 | 24.00 | 288.00 | 19.00 | 280.00 |
| | 5 | 21.00 | 33.25 | 399.00 | | | | | | |
| | 10 | 42.00 | 31.50 | 378.00 | | | | | | |
| | 15 | 63.00 | 29.75 | 357.00 | | | | | | |
| | 20 | 84.00 | 28.00 | 336.00 | | | | | | |
| | 25 | 105.00 | 26.25 | 315.00 | | | | | | |

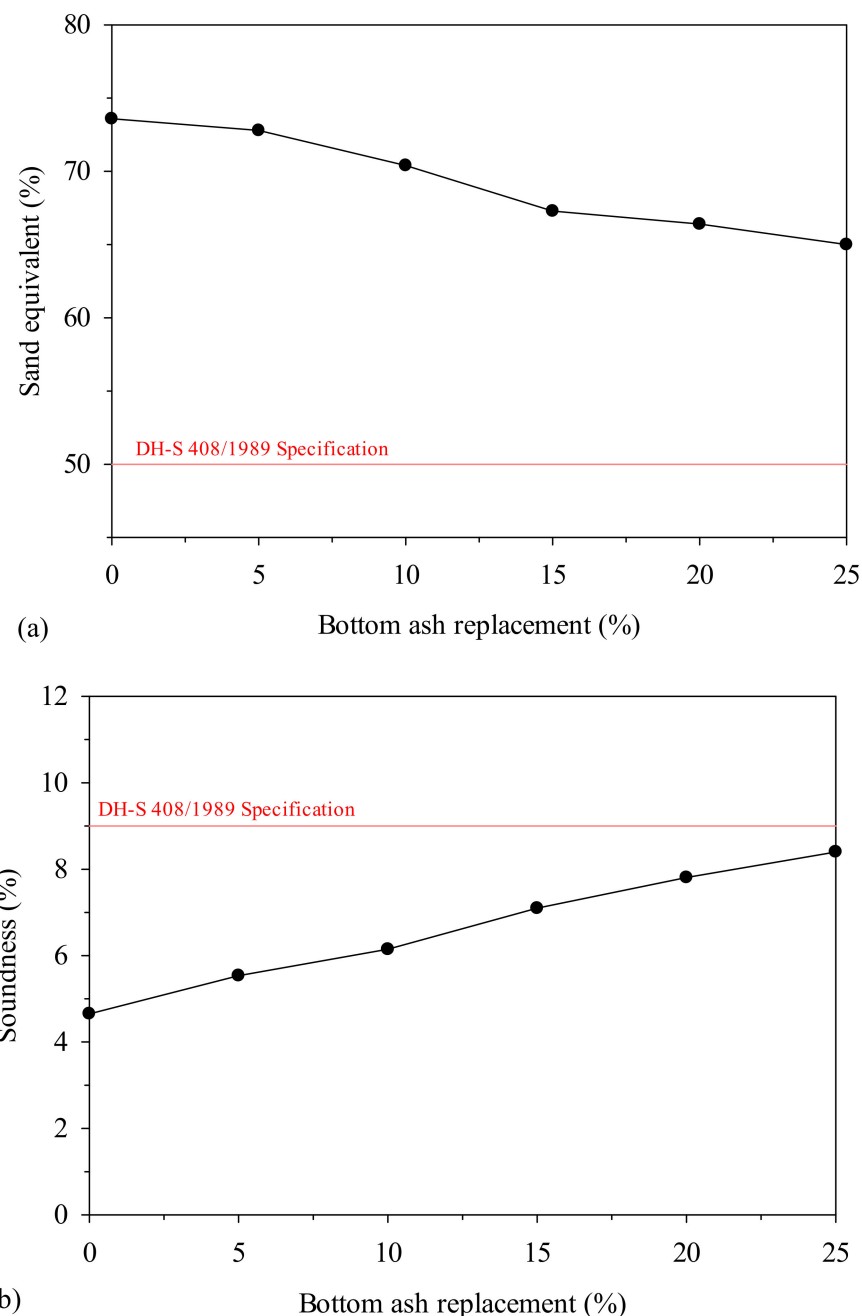

**Figure 4.** Relationship of (**a**) sand equivalent versus BA-replacement ratio and (**b**) soundness versus BA-replacement ratio, of BA-asphalt concretes.

Bituminous binder was heated to the target temperature at 160 °C. The combined aggregates, at different BA-replacement ratios, were oven dried at 180 °C. The combined aggregates and bituminous binder were, then, mixed within 60 s at 150 °C, before compaction. The asphalt concretes with different BA-replacement ratios were prepared at 4% air void, based on DH-T 604/1971 [51], in the standard 101.60 mm metallic mold, using the Marshall method with 75 blows.

### 3.2. Marshall Stability and Flow

Marshall stability and flow of asphalt concretes were evaluated, by using a compression machine, with the automatic data recorder at the deformation rate of 50 mm/min, according to ASTM D 6927 [52]. The deformation throughout the test was measured, using a linear variable differential transformer (LVDT) connecting with the related software.

### 3.3. Strength Index

The asphalt concretes were divided into two groups, with three samples for each group. In the first group, the asphalt concrete samples were soaked in sodium chloride, with 5 g per 0.001 m$^3$, for 24 h at 60 °C, in a controlled-temperature water bath, then soaked in 25 °C water for another hour, before the stability test. The second group of samples was unsoaked. The strength-index value is the stability ratio of the soaked asphalt concrete samples to the unsoaked asphalt concrete samples, in accordance with DH-S 413/2001 [53].

### 3.4. Indirect Tensile Strength

Indirect tensile strength (ITS) was carried out in accordance with ASTM D6931 [54]. The samples were installed in a loading frame, equipped with a loading strip of 19 mm wide and 125 mm long, in a controlled-temperature chamber. To evaluate the effect of temperature on ITS, the samples were subjected to the target temperatures of 25 °C, 40 °C, and 50 °C, throughout the test. The samples were subjected to vertical loading, with a deformation rate of 50 mm/min, by using a compression machine. The load and deformation were, automatically, recorded by the related software. The ITS is calculated, based on the following equation:

$$ITS = \frac{2P}{\pi dt} \tag{1}$$

where $P$ is the maximum load (N), $t$ is the thickness of the sample (mm), and $d$ is the diameter of the sample (mm).

### 3.5. Indirect Tensile Resilient Modulus (IT $M_r$)

The resilient property is, commonly, evaluated on the basis of the indirect tensile loading [55], for asphalt-concrete design. The sample was subjected to a haversine-loading pulse, with a loading frequency of 1.0 Hz, at a loading period of 0.1 s and a rest period of 0.9 s, at 25 °C. The stress level was imposed at 15% of ITS, as recommended by ASTM D4123 [54]. The repeated tensile stress was applied until 200 cycles, and the horizontal deformation for each loading cycle was, automatically, measured by an LVDT. The IT $M_r$ is calculated, according to the following equation:

$$IT\ Mr = \frac{P(v + 0.27)}{t\Delta h} \tag{2}$$

where $IT\ M_r$ is the indirect tensile resilient modulus (MPa), $P$ is the applied repeated load (N), $v$ is the Poisson's ratio (assumed to be 0.35, according to Austroads (2004) [38], $\Delta h$ is the recoverable horizontal deformation (mm), and $t$ is the sample thickness (mm).

### 3.6. Indirect Tensile Fatigue (ITF)

The ITF test on BA-asphalt concretes was carried out, in accordance with EN 12697-24 [56]. The haversine load pulse, at a loading frequency of 1.0 Hz (a loading period of 0.1 s and a rest period of 0.9 s), at stress levels of 250 kPa, 300 kPa, and 350 kPa, was selected. In order to determine the load and deformation behavior of BA-asphalt concrete, the vertical deformation was, automatically, measured using an LVDT, for each loading cycle. The repeated loading was applied, until the sample completely failed (vertical deformation larger than 12 mm).

### 3.7. Permanent Deformation

Permanent deformation of BA-asphalt concretes was measured via the dynamic creep test, in accordance with AS 2891.12.1 (1995) [57]. The sample was subjected to a static vertical stress of 10 kPa for 30 s, prior to the test. Thereafter, the square-wave loading pulse, at a loading frequency of 0.5 Hz with a loading period of 1.0 s and a rest period of 1.0 s, was applied on the sample. The repeated stress was applied at 50 °C, until 1800 pulses. The permanent deformation was determined as the accumulation of irrecoverable deformation, for each loading pulse.

### 3.8. Rutting Resistance

The rutting resistance test on asphalt concretes was carried out, in accordance with AASHTO T 324 [58], by using a Hamburg wheel-tracker testing machine. The sample dimension was 150 mm in diameter and 60 mm in thickness. A moving load of 1.5 kN (equivalent to standard tire pressure of 707 kPa) was applied on the sample surface, through a steel wheel with a diameter of 203 mm and a width of 47 mm, moving back and forth on asphalt concretes, at 50 °C.

### 3.9. Skid Resistance

The skid resistance test on asphalt concretes was performed, based on ASTM E 303, [59] using the British pendulum tester. The skid resistance value was determined by the friction from the swing of the rubber platform touching the sample surface and is expressed as British pendulum number, BPN. In order to assess the surface friction after various traffic loading, the BPN of the Hamburg wheel tracker samples was measured after 0, 2000, 4000, 6000, and 6000 wheel cycles (moving back and forth).

## 4. Results and Discussion

### 4.1. Marshall Properties of BA-Asphalt Concretes

The basic properties of BA-asphalt concretes, at different BA replacement ratios, are summarized in Table 5. Due to high porosity, BA was found to have a lower specific gravity and specific surface than the natural fine aggregate. The higher specific surface results in a higher bituminous binder absorption. Hence, the higher BA replacement ratio yields higher optimum bituminous binder content [27]. As a result of higher optimum bituminous binder, the voids in mineral aggregate (VMA) and voids filled with bitumen (VFB) increase, with the increased BA replacement ratio at the same air void (4%). Consequently, the higher VMA results in a lower density at a higher BA-replacement ratio.

**Table 5.** Properties of BA-asphalt concretes.

| BA Replacement-Ratio by wt. of Aggregate (%) | Optimum Bituminous Binder (%) | Density (%) | VMA (%) | Air Voids (%) | VFB (%) |
|---|---|---|---|---|---|
| 0 | 5.0 | 2.391 | 14.3 | 4.0 | 81.3 |
| 0 | 4.9 | 2.389 | 14.7 | 4.0 | 75.5 |
| 5 | 5.0 | 2.385 | 15.0 | 4.0 | 75.8 |
| 10 | 5.1 | 2.379 | 15.7 | 4.0 | 75.84 |
| 15 | 5.2 | 2.368 | 16.5 | 4.0 | 76.4 |
| 20 | 5.4 | 2.356 | 17.8 | 4.0 | 76.7 |
| 25 | 5.5 | 2.338 | 18.7 | 4.0 | 78.4 |

Figure 5 presents the relationship between Marshall stability and the BA-replacement ratio. The solid line represents the minimum requirement of 8 kN, specified by DH-S 408/1989. BA-asphalt concrete, at various replacement ratios, met the minimum requirement. The stability of BA-asphalt concretes is 8.9 kN, 9.6 kN, 9.5 kN, 9.5 kN, 9.4 kN, and 9.2 kN for 0%, 5%, 10%, 15%, 20%, and 25% BA-replacement ratios, respectively. The highest stability is found at the optimum BA-replacement ratio (OPT-BA), of 5%. Beyond the OPT-BA, stability decreases with any further increase in the BA-replacement ratio. BA-asphalt concrete has higher stability than asphalt concrete, even at the highest BA-replacement ratio tested, of 25%.

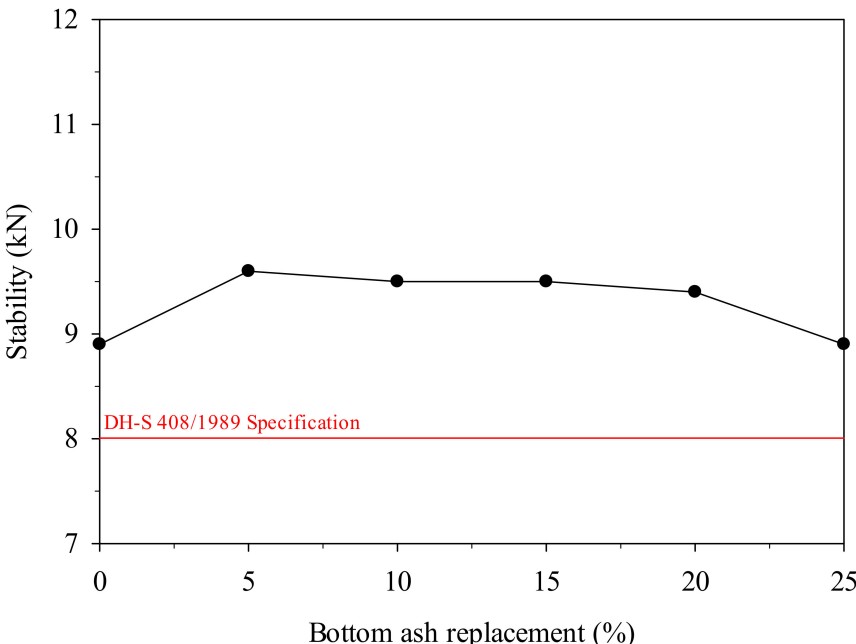

**Figure 5.** Relationship between stability and BA-replacement ratio of BA-asphalt concretes.

Luo et al. (2017) [27] revealed that the rough surface of BA could increase the thickness of the asphalt film, due to its lipophilic reaction. This thick asphalt film is the influence factor affecting the bonding strength between the aggregate particles of asphalt concrete; therefore, it strengthens the stability of asphalt concrete. However, the excess BA content causes ineffective asphalt film, resulting in a larger VMA and, ultimately, decreased stability.

The Marshall flow of BA-asphalt concretes is illustrated in Figure 6. The solid line represents the flow value, specified by DH-S 408/1989 (between 2.29 mm and 4.32 mm). The increased BA-replacement ratio causes an increase in flow. The highest flow is found at the 25% BA-replacement ratio, due to the highest optimum bituminous binder content. The flow of BA-asphalt concretes is 3.22 mm, 3.28 mm, 3.33 mm, 3.34 mm, 3.39 mm, and 3.43 mm, for 0%, 5%, 10%, 15%, 20%, and 25% BA-replacement ratios, respectively.

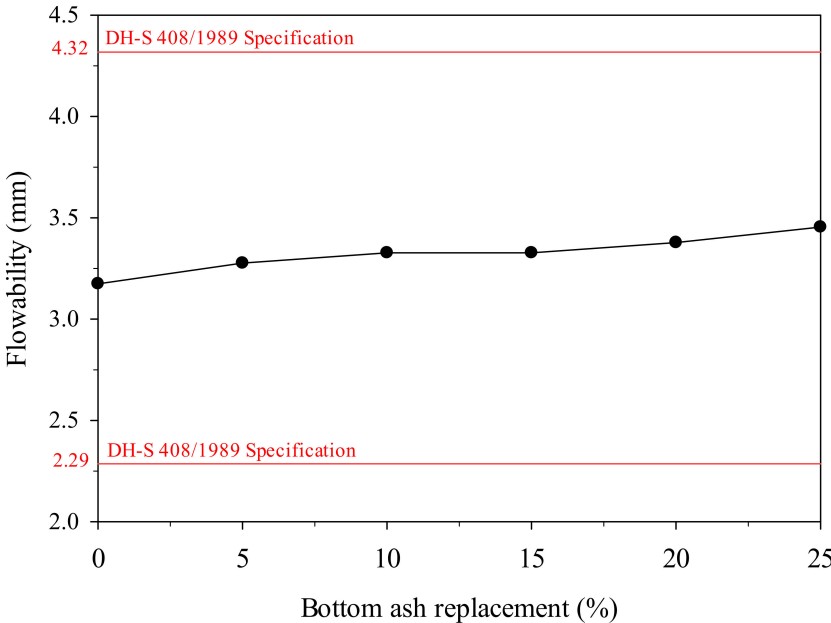

**Figure 6.** Relationship between Marshall flow and BA-replacement ratio of BA-asphalt concretes.

The relationship between strength index versus BA-replacement ratio is shown in Figure 7. The strength index of BA-asphalt concretes is higher than the minimum requirement of DH-S 408/1989, for all replacement ratios. The highest strength index is found at the OPT-BA, of 5%. The highest strength index of BA-asphalt concretes is 94.7%, while it is 93.0% for asphalt concrete. Similar to stability results, the strength index of BA-asphalt concrete decreases, with a further increase in the BA-replacement ratio, beyond OPT-BA. Therefore, it is interpreted that the asphalt film in BA-asphalt concrete, at optimum asphalt content, results in an improvement of resistance to deterioration against the sodium chloride attack.

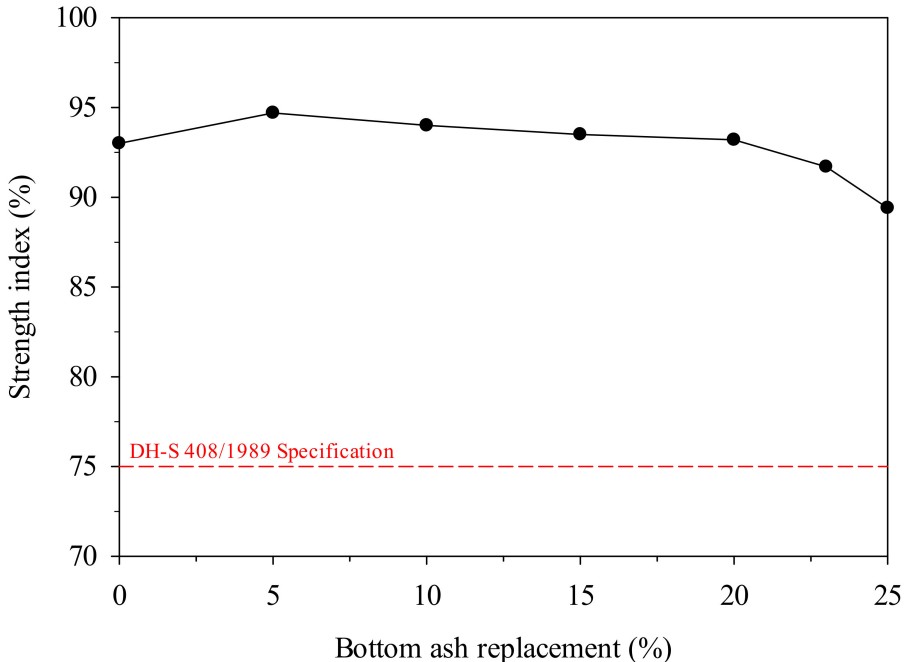

**Figure 7.** Relationship between strength index and BA-replacement ratio of BA-asphalt concretes.

## 4.2. Performance of BA-Asphalt Concretes

The relationship between ITS and the BA-replacement ratio, under different temperatures, is illustrated in Figure 8. At the same BA-replacement ratio, the elevated temperature causes the decrease in ITS, due to the softening of the asphalt binder, especially at 60 °C. BA particles are rougher than the natural fine particles, hence they have a higher interlocking ability [27]. As a result, the BA replacement contributes to the ITS improvement. The highest ITS at any temperatures is found at the OPT-BA, of 5%. For instance, at 25 °C, the ITS of BA-asphalt concretes is 0.244 MPa, 0.250 MPa, 0.249 MPa, 0.247 MPa, 0.240 MPa, and 0.220 MPa, for 0%, 5%, 10%, 15%, 20%, and 25% BA-replacement ratios, respectively.

To investigate the resistance to ITS reduction against the elevated temperature, the relationship between $ITS_{\Delta t}/ITS_{25}$ °C and temperature was developed and is demonstrated in Figure 9, where $ITS_{\Delta t}$ is the strength at any temperature, and $ITS_{25}$ °C is the initial ITS at 25 °C. The higher $ITS_{\Delta t}/ITS_{25}$ °C at the same temperature indicates a higher resistance to strength degradation, due to the raised temperature. As expected, the highest $ITS_{\Delta t}/ITS_{25}$ °C value is found at the OPT-BA. For example, the $ITS_{\Delta t}/ITS_{25}$ °C value at 60 °C is 0.2 and 0.4, for 0% and 5% BA-replacement ratios, respectively. The relationship for BA-asphalt concretes lies above that of asphalt concrete, except for the 25% BA-replacement ratio. This implies that the suitable BA-replacement ratio has a superior resistance to ITS reduction, due to the raised temperature.

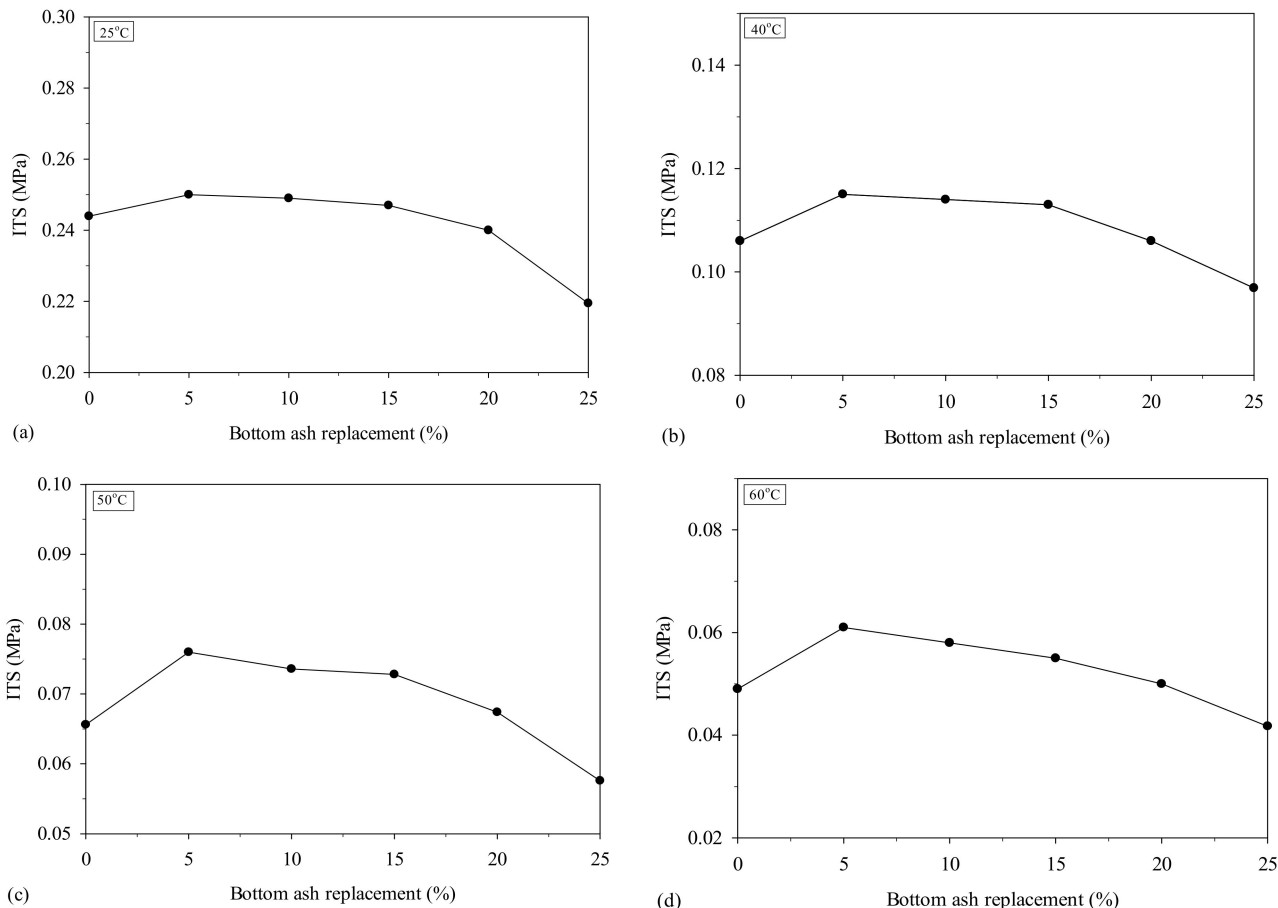

**Figure 8.** Relationship between ITS and BA-replacement ratios of BA-asphalt concretes, under temperatures of (**a**) 25 °C, (**b**) 40 °C, (**c**) 50 °C, and (**d**) 60 °C, respectively.

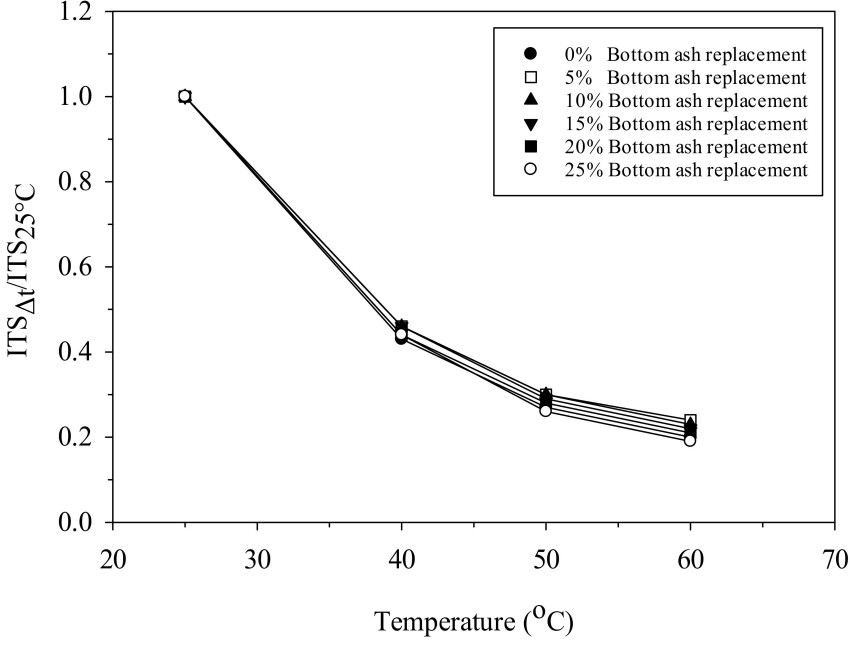

**Figure 9.** Relationship between $ITS_{\Delta t}/ITS_{25}$ °C and temperature of BA-asphalt concretes, at different BA-replacement ratios.

Figure 10 presents the relationship between indirect tensile resilient modulus (IT $M_r$) and the BA-replacement ratio. It is evident that the BA-replacement ratio enhances the IT $M_r$ of asphalt concretes. The highest IT $M_r$ is found at the same OPT-BA, of 5%, which is 4.42% higher than the IT $M_r$ of the asphalt concrete. When the BA -eplacement ratios are greater 10%, the IT $M_r$ of BA-asphalt concretes is found to be lower than that of the asphalt concrete.

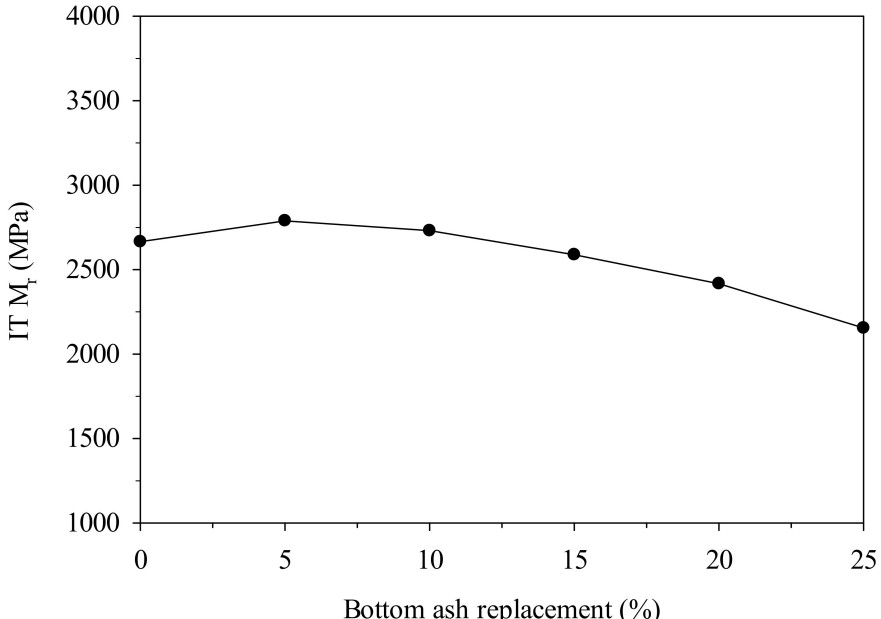

**Figure 10.** Relationship between IT $M_r$ and BA-replacement ratio.

Both improved IT $M_r$ and ITS are, directly, related to the BA-replacement ratio. As such, the relationship between IT $M_r$ and ITS of BA-asphalt concrete could be developed and is demonstrated, in Figure 11, as a linear function for all BA-replacement ratios. In other words, improved ITS leads to an increase in the resistance to plastic deformation, under repeated tensile stress. With the high coefficient of determination, the IT $M_r$ of BA-asphalt concretes can be estimated, when the ITS is known.

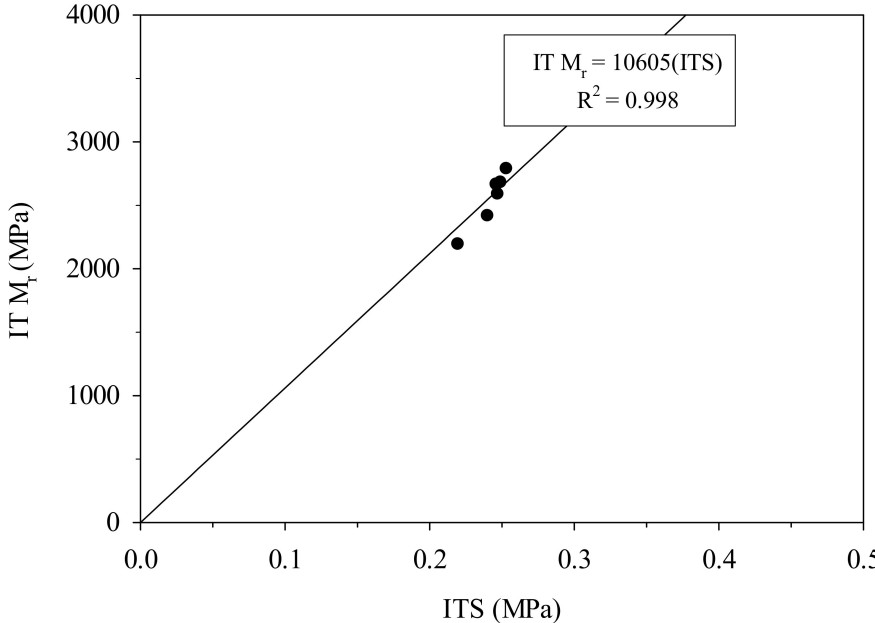

**Figure 11.** Relationship between IT $M_r$ and ITS of BA-asphalt concretes, at different BA-replacement ratios.

Figure 12 presents the relationship between deformation and the number of cycles (*N*) of BA-asphalt concretes, under the ITF test, at the three applied stress levels of 250 kPa, 300 kPa, and 350 kPa. The deformation behavior can be categorized into three distinct zones, as demonstrated in Figure 12a. In the first zone, the deformation increases rapidly, with a small initial *N*, due to the decrease in air void. The aggregates in asphalt concretes are, thereafter, closer together, and the deformation slightly increases with an increased *N*, in the linear function in the second zone. In this zone, the micro-cracks, gradually, generate and, then, propagate to macro-cracks, with a further increase in *N*. The macro-cracks, ultimately, lead to fatigue failure in the third zone. The indirect tensile fatigue life (ITFL) is defined as the intersection point between the tangent lines, projected from the second and third zones (Figure 12a) [25,39].

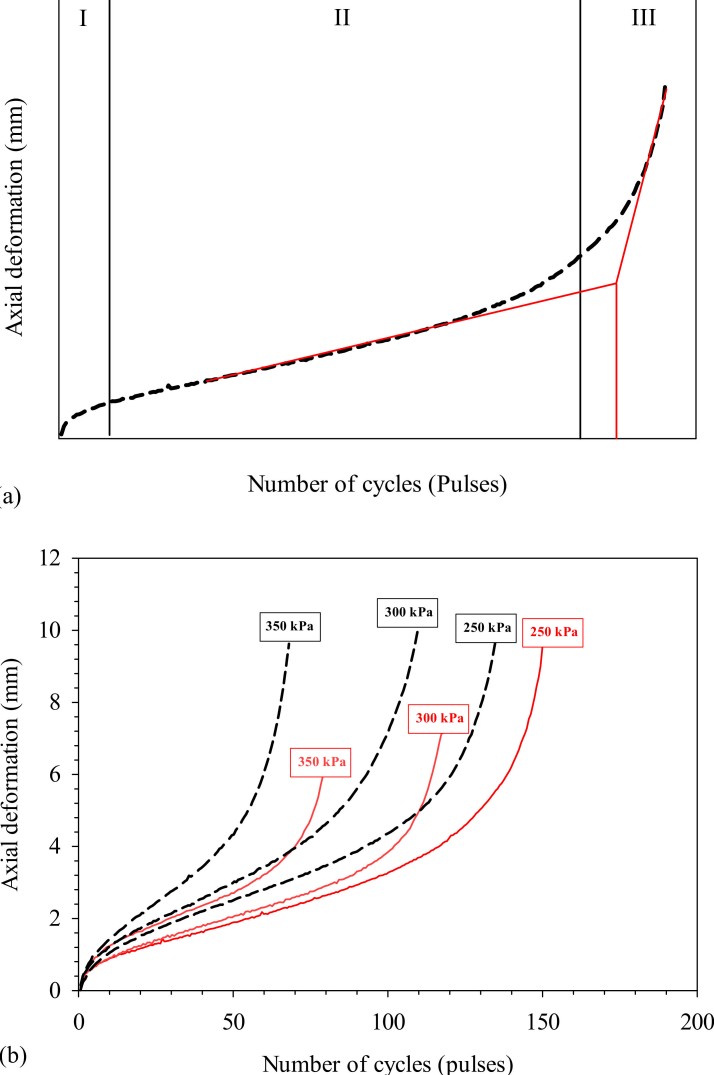

**Figure 12.** Relationship between axial deformation and number of cycles, under indirect tensile fatigue test; (**a**) schematic plot and (**b**) BA-asphalt concretes, at different BA-replacement ratios and stress levels.

The influence of stress level on deformation behavior is illustrated in Figure 12b. The lower applied stress results in a larger deformation, at the transition between zone 1 and zone 2, for both conventional and BA-asphalt concretes. The transition between zone 1 and zone 2 is found at a similar *N*, for the three applied stresses studied. The higher applied stress causes a lower service life, which can be observed by the decreased span of zone 2 and a decrease in ITFL. It is noted that the three studied applied stresses were higher than

the ITS of asphalt and BA-asphalt. Hanoon et al. (2016) [60] revealed that asphalt concretes can carry cyclic vertical stress higher than ITS, due to the difference in strain. Initially, the cyclic strain, due to the repeated load, was much lower than the static failure strain at ITS.

At the same *N*, BA-asphalt concretes exhibit a lower total deformation, in zone 2, than the asphalt concretes for all stress levels. The lowest deformation is found at the OPT-BA. The deformation increases with a further increase in BA-replacement ratio, beyond the OPT-BA. The lower deformation in zone 2 of BA-asphalt concretes is associated with a higher ITFL, as illustrated in Figure 12b.

The relationship between ITFL and BA-eplacement ratio, under different stress levels, is summarized in Figure 13a. Similar to the ITS and IT $M_r$ results, the highest ITFL at various stress levels is found at the OPT-BA, for all stress levels. The ITFL of BA-asphalt concretes is 150 pulses, 118 pulses, and 79 pulses at OPT-BA, while the ITFL of the asphalt concretes is 135 pulses, 110 pulses, and 68 pulses, for stress levels of 250 kPa, 300 kPa, and 350 kPa, respectively. The ITFL decreases with a further increase in BA, beyond the OPT-BA, especially the 25% BA-replacement ratio. The ITFL of the 25% BA-replacement ratio is lower than that of asphalt concrete, which is 130 pulses, 97 pulses, and 57 pulses, for stress levels of 250 kPa, 300 kPa, and 350 kPa, respectively.

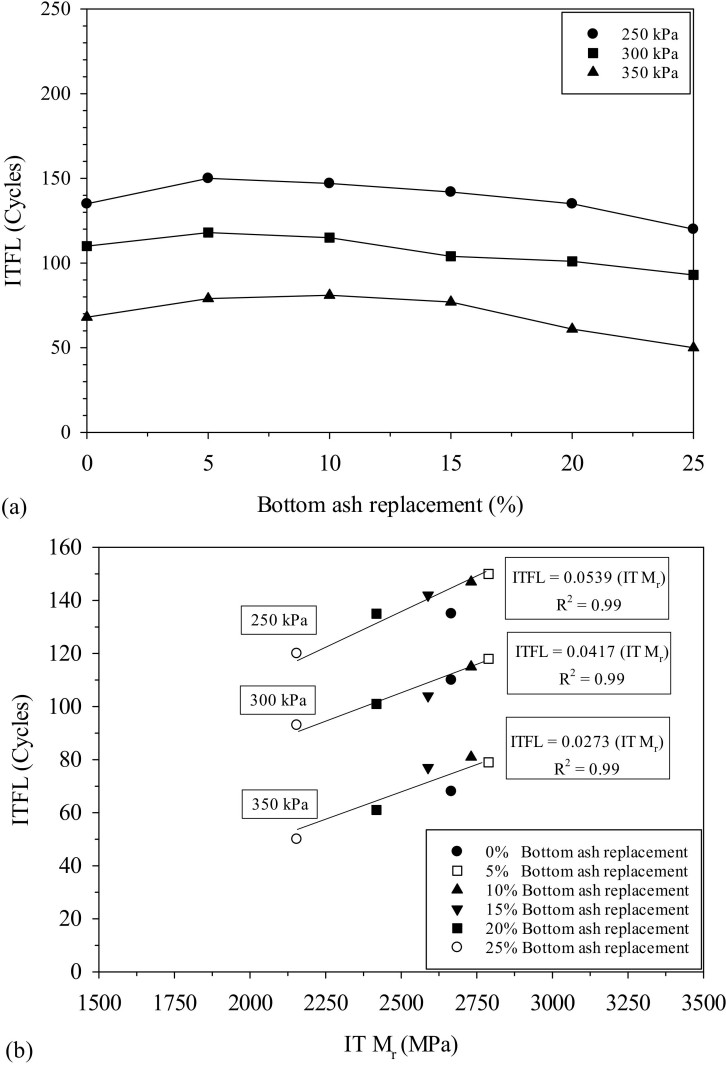

(a)

(b)

**Figure 13.** (**a**) Relationship between ITFL and BA-replacement ratio of BA-asphalt concretes, at different stress levels, and (**b**) relationship between the ITFL and IT Mr of BA-asphalt concretes at different BA-replacement ratios under different stress levels.



Typically, the higher IT $M_r$ is associated with a higher fatigue life for the pavement base course. The relationship between the ITFL and IT $M_r$ of BA-asphalt concretes, at different BA-replacement ratios under different stress levels, was plotted and is illustrated in Figure 13b. It is evident that the IT $M_r$ is linearly correlated to ITFL, for both asphalt concretes and BA-asphalt concretes, at a specific stress level. At the same IT $M_r$, the higher stress level results in a lower ITFL, causing a lower slope of the relationship.

The dynamic creep and the Hamburg wheel tracker tests on asphalt concretes were performed on BA-asphalt concretes, to investigate the role of the BA-replacement ratio on the rut resistance. The relationship of permanent deformation (*PD*) versus the number of cycles under the dynamic creep test ($N_P$) of BA-asphalt concretes at the different BA-replacement ratios is shown in Figure 14. The relationship pattern for BA-asphalt and asphalt concretes, is essentially similar. At the initial state ($N_P < 200$ pulses), the deformation increases significantly, with a small $N_P$, due to compression of the air void. The deformation increases gradually, in proportion to $N_P$, until the end of the test at 1800 cycles. As expected, the lowest *PD* is found at the OPT-BA, while the highest *PD* is found at the 25% BA-replacement ratio. The *PD* of BA-asphalt concretes is 2750 micro strains, 2680 micro strains, and 4455 micro strains, for 0%, 5% and 25% BA-replacement ratios, respectively.

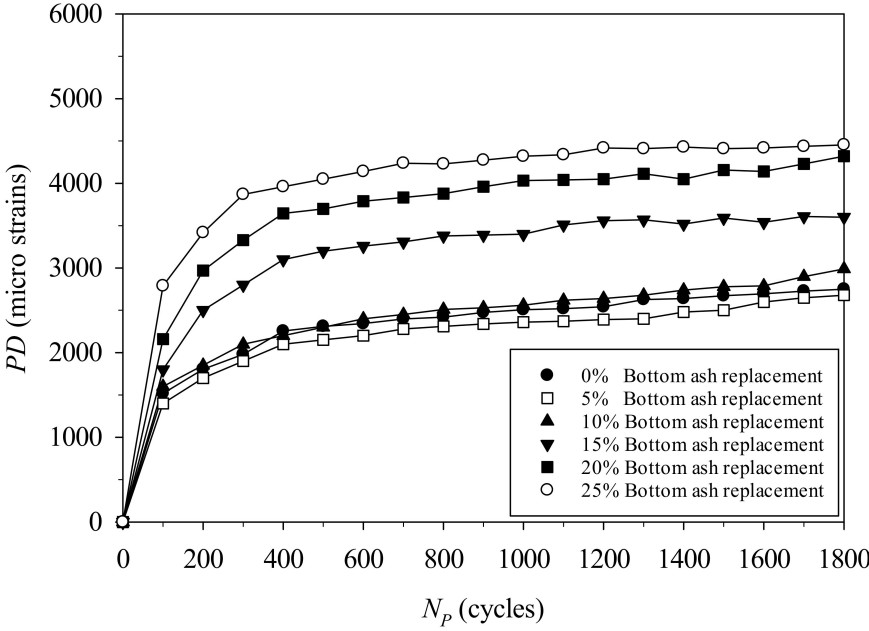

**Figure 14.** Relationship between permanent deformation and number of cycles, under dynamic creep test, at different BA-replacements ratios.

Figure 15 presents the relationship between rut depth and the number of wheel cycles ($N_W$), under the wheel-tracker test. The schematic plot of the typical wheel-tracker-test result is demonstrated in Figure 15a. The relationship between rut depth and $N_W$ is divided into two major zones, based on its slopes. In the first slope, designated as creep slope, the rut depth linearly increases with $N_W$, due to the creep of asphalt concretes under repeated stress. In the second slope (stripping slope), the slope of the relationship indicates the degree of damage of asphalt concretes, due to the effect of moisture and temperature. The higher slope of zone 2 is associated with a higher rate of increased rut depth with the number of wheel cycles, due to stripping in asphalt concretes. The intersection between creep slope and stripping slope is defined as the inflection point, which is the final state of creeping and initial state of stripping [61]. The results of BA-asphalt concretes, at various replacement ratios, are presented in Figure 15b. The highest resistance to rut depth is found at the OPT-BA.

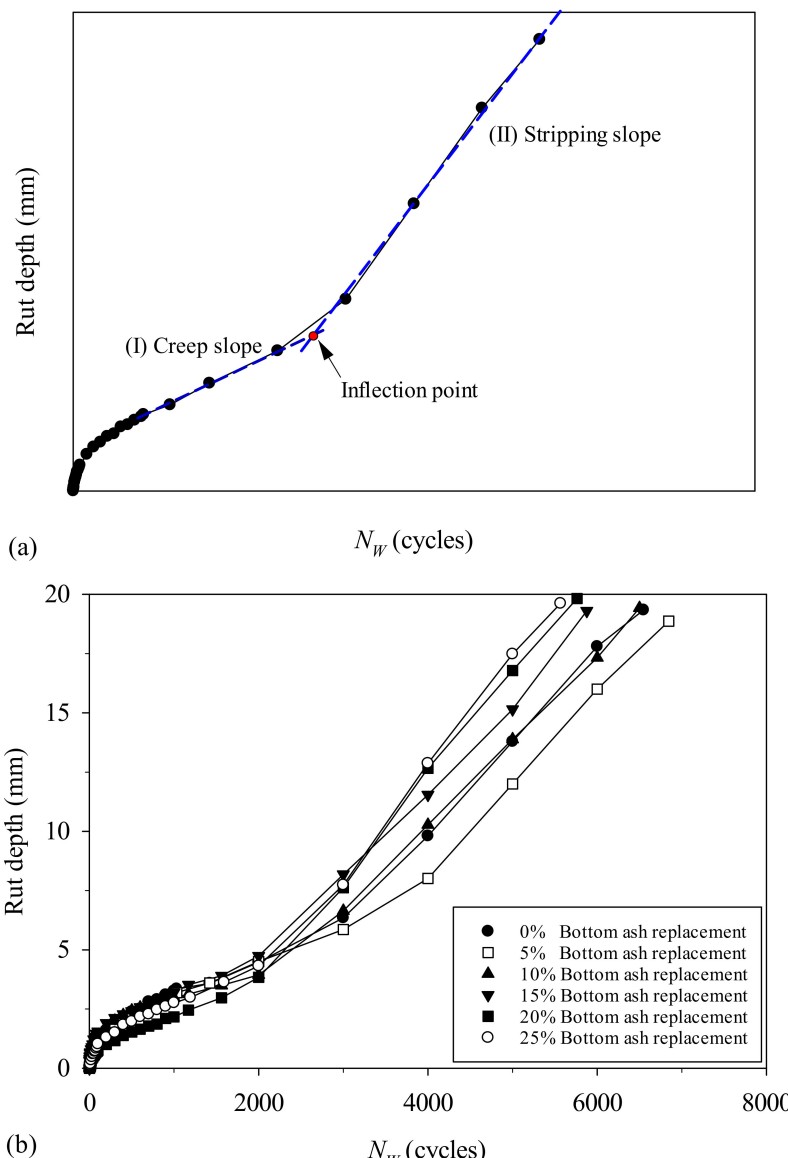

**Figure 15.** Relationship between rut depth and $N_w$ under wheel-tracker test; (**a**) schematic plot and (**b**) BA-asphalt concretes, at different BA-replacement ratios.

Theoretically, the asphalt concretes with a lower creep and higher inflection point have a higher rut resistance [60]. The relationships of creep slope, inflection point, and stripping slope with the BA-replacement ratio are demonstrated in Figure 16a–c. The creep slope of BA-asphalt concretes is lower than that of asphalt concrete, due to the superior resistance to permanent deformation (Figure 16a). The lowest creep slope is found at the OPT-BA. The creep slope of BA-asphalt concrete is 0.00212 mm/$N_w$, 0.00154 mm/$N_w$, 0.00168 mm/$N_w$, 0.001745 mm/$N_w$, 0.00185 mm/$N_w$, and 0.00205 mm/$N_w$ for 0%, 5%, 10%, 15%, 20%, and 25% BA-replacement ratios, respectively. BA replacement, also, improves the resistance to stripping slope and the inflection point (Figure 16b). The inflection point of asphalt concretes increases with the increased BA-replacement ratio, up to the highest value, at the OPT-BA. BA replacement reduces the stripping slope benchmarked to bituminous binder, except the 25% BA-replacement ratio (Figure 16c).

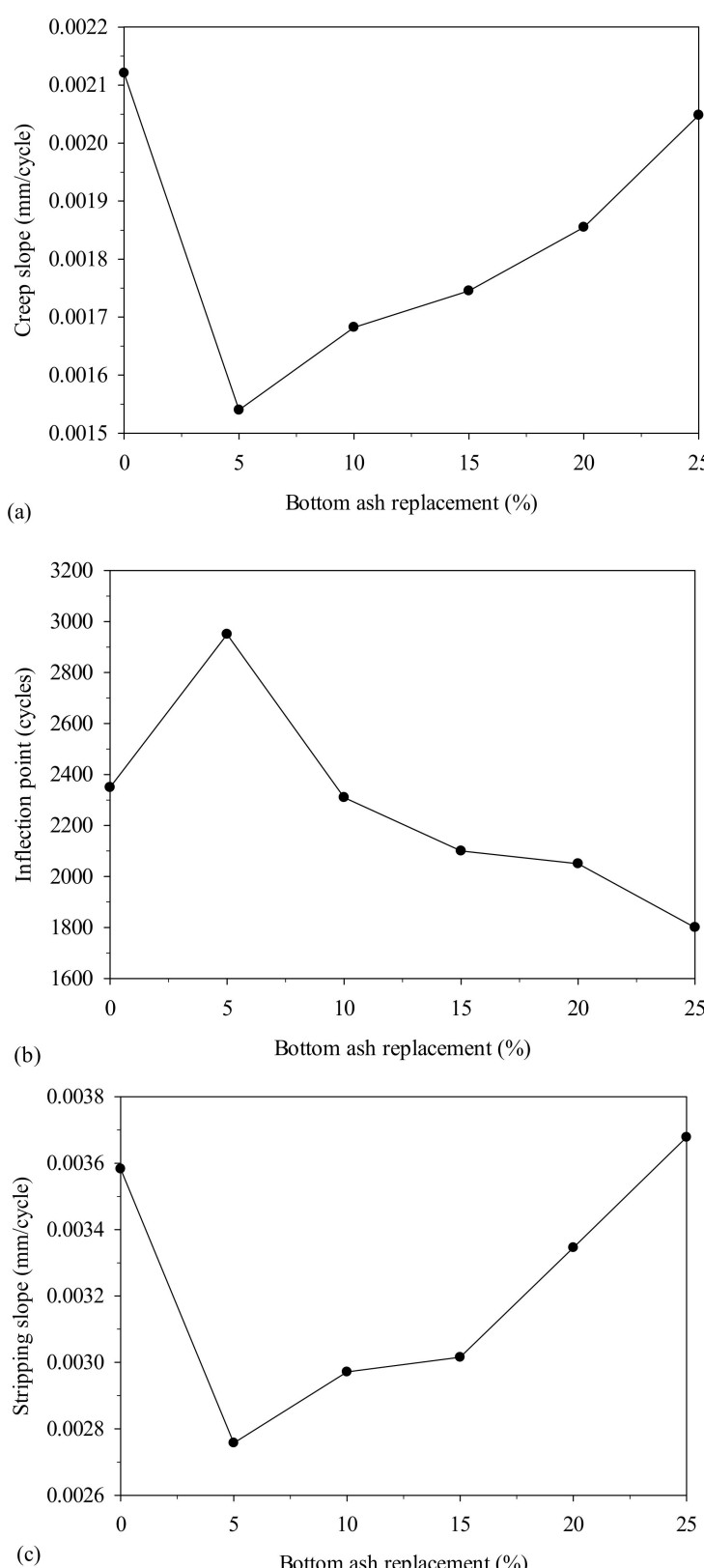

**Figure 16.** Relationship of (**a**) creep slope, (**b**) inflection point, and (**c**) stripping slope with BA-replacement ratio of BA-asphalt concretes, under wheel-tracker test.

From the analysis of the test results, it is evident that IT $M_r$ is, directly, related to the compressive resilient modulus, which controls permanent deformation and rutting resistance. A higher IT $M_r$ is associated with a lower *PD* and rut depth. Figure 17 illustrates

the linear relationship between IT $M_r$ and *PD*. In other words, the OPT-BA improved both the tensile and compressive cyclic performance, leading to improved indirect tensile fatigue life and rutting resistance.

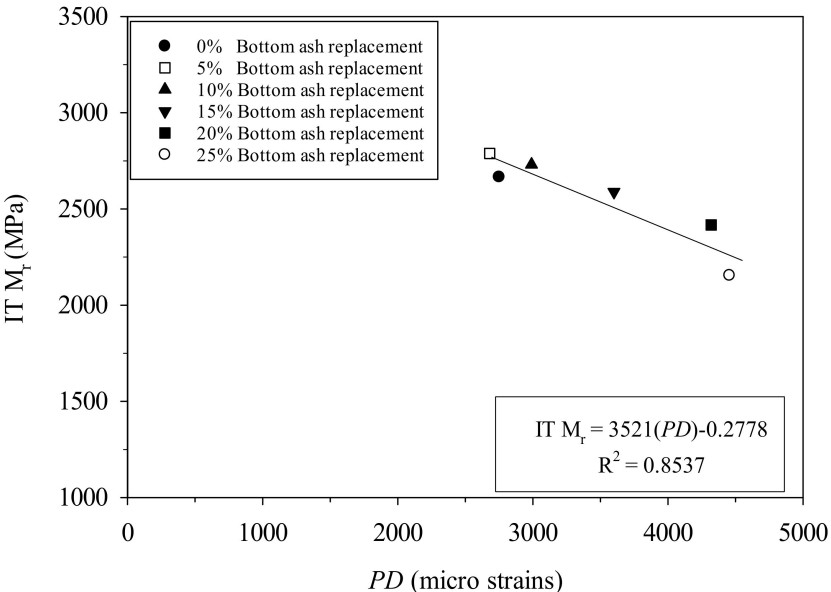

**Figure 17.** Relationship between IT $M_r$ and permanent deformation of BA-asphalt concretes, at different BA-replacements ratios.

The relationship of the British pendulum number (BPN) and the BA-replacement ratio, at different number of wheel cycles ($N_W$), is presented in Figure 18. The higher $N_W$ significantly causes lower BPN, for all the BA-replacement ratios tested. At the same $N_W$, the increased BA-replacement ratio contributes to an increased BPN. The BPN at $N_W = 4000$ cycles can be determined only at 0%, 5%, 10%, 15%, and 20% BA replacement, since the sample at the 25% BA-replacement ratio failed, before $N_W = 4000$ cycles. The highest skid resistance is found at the OPT-BA, for all studied $N_W$. The improved skid resistance of BA-asphalt concrete contributes to the improvement of safe driving performance.

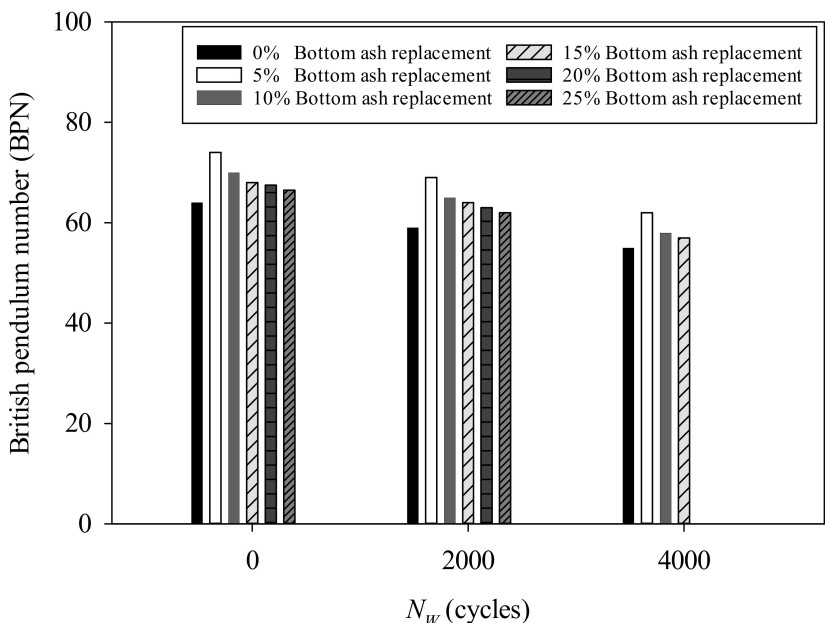

**Figure 18.** Relationship between British pendulum number and $N_W$ of BA-asphalt concretes, at different BA-replacement ratios.

## 5. Conclusions

This research evaluated the performance improvement of asphalt concretes, using BA replacement. The following significant findings can be summarized from this research:

1.  With a higher specific surface, the higher BA-replacement ratio yields higher optimum bituminous binder content, at the same air void (4%). BA replacement could increase the thickness of asphalt film, due to its lipophilic reaction. This thick asphalt film is the influence factor, affecting the mechanical strength of asphalt concrete. The highest Marshall stability, strength index, and ITS were found at the OPT-BA, of 5%. Beyond 5% BA replacement, the excessive thick film on BA causes a weak zone and, hence, a reduction in mechanical properties.

2.  The improved ITS led to an increase in the resistance to plastic deformation, under repeated tensile stress. Hence, IT $M_r$ was improved, and the highest IT $M_r$ was, also, found at the OPT-BA. At the same number of repeated tensile loadings, BA-asphalt concretes exhibited a lower total deformation than the asphalt concretes, for all stress levels. Similar to ITS and IT $M_r$ results, the highest ITFL was found at the OPT-BA, of 5%, for all stress levels tested. IT $M_r$ is linearly correlated to ITFL, for both asphalt concretes and BA-asphalt concretes, at a specific stress level.

3.  The wheel-tracker-test results indicated that the BA replacement could improve the creep slope, inflection point, and stripped slope of asphalt concretes, which were in agreement with the dynamic creep test, in that it could, also, improve the *PD*. As such, BA-asphalt concretes at the OPT-BA had a lower rut depth than the asphalt concretes, at the same number of wheel cycles.

4.  Based on analysis of the performance tests, IT $M_r$ was, directly, related to the compressive resilient modulus, which controls permanent deformation and rutting resistance. A higher IT $M_r$ is associated with lower *PD* and rut depth. In other words, the OPT-BA improved both the tensile and compressive cyclic performance, leading to improved ITFL and rutting resistance.

5.  BPN for all BA-replacement ratios tested decreased with the increased number of wheel cycles. BA replacement was able to improve the BPN, at a given number of wheel cycles, and the highest BPN is found at the OPT-BA, of 5%.

6.  With the Sustainable Development Goals (SDGs), including the three pillars of sustainability (society, economy, and environment) for the government of Thailand, the usage of BA replacement, even at a small amount, of 5% (given the long road network in Thailand), would benefit the requirements for all three pillars. The outcome of this research will promote the usage of BA, a by-product from coal-fired powerplants, as a cleaner additive in sustainable pavement application. This usage will reduce the environmental problems and create a value-added by-product. This research will be useful for national and international road authorities as well as powerplant administrators worldwide.

**Author Contributions:** Conceptualization, A.S. and S.H.; methodology, A.S. and S.H.; software, K.A.; validation, K.A. and A.S.; formal analysis, A.S., K.A. and T.Y.; investigation, A.S., K.A. and T.Y.; resources, A.S. and C.B.; writing—original draft preparation, A.S.; writing—review and editing, A.B., A.S., S.H., K.A., T.Y., M.H. and A.A.; visualization, A.S.; supervision, A.S. and S.H.; project administration, A.S., S.H. and C.B.; funding acquisition, A.S. and S.H. All authors have read and agreed to the published version of the manuscript.

**Funding:** Electricity Generating Authority of Thailand (Grant Number 63-N001000-11-IO.SS03N3008558) and National Science and Technology Development Agency (Grant Number P-19-5203).

**Institutional Review Board Statement:** Not applicable.

**Informed Consent Statement:** Not applicable.

**Data Availability Statement:** The data presented in this study are available on request from the corresponding author.

**Acknowledgments:** This research was financially support by the Electricity Generating Authority of Thailand (Grant Number 63-N001000-11-IO.SS03N3008558). The authors, also, appreciate the support from National Science and Technology Development Agency, under the Chair Professor Program (Grant Number P-19-52303).

**Conflicts of Interest:** The authors declare no conflict of interest.

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
