# Peer review of "Improved Performance of Asphalt Concretes using Bottom Ash as an Alternative Aggregate"

_sustainability, doi:10.3390/su14127033_

Round 1
Reviewer 1 Report
Dear authors,
Thank you for your article, however it needs some corrections to make more pleasant for readers.
- Abstract: The abstract needs to be rewritten and spelled checked. It is obvious, it was copied from different format and contains hyphens where they should not be. Also there is mentioned variant with 30 % BA, which is not included in the paper.
- Introduction: The literature review is quite fine, but you mostly reference to papers 15 – 20 years old or even older. There are no more recent papers on this topic? If not, why is it, isn’t it because the previous paper showed some problems with this solution? Please add more recent research papers on this topic.
- Asphalt cement – instead of asphalt cement please use the more used terminology “bituminous binder”.
- Specify clearly what percentage of BA means. It is in text several times, but sometimes it is defined so poorly, it is misleading. If I understood clearly, it is x % of mass of fine aggregate. Please add the recipe for mixtures, so readers can understand the total amount of BA in the mixture.
- I´m very sorry but I don’t understand the parameters explain in paragraph from line 290 (+ Figure 9), please can you explain in more details, why the ration is calculated and what are the effects of test result?
- The conclusion has to be rewritten. It included only 12 lines. There is no explanation why the lowest BA content reached the best values. What it means for asphalt plant to add such a small amount of another additive? Is it economically favourable for total price of asphalt mixture? Is it really ecological favourable, if only such a small amount will be added (in order to BA transportation, storaging etc.). Does it really make any sense to use only 5 % BA in the mixture? What can be add for improving the properties with higher amount of BA in mixture?
- Editorly: Be careful with declination of unites, sometimes you decline, sometimes not.
- The annex with figure and tables have to re-formatted! The format is terrible and it shows that no one checked it before the uploading.
Reviewer 2 Report
No comment
Author Response
The authors are thankful for recognizing the significance of this research.

Reviewer 3 Report
- You should improve the language of the manuscript. You are using some weird words. exp.:"in infrastructure works is in infancy due to limited knowledge an"
- The abstract should be improved. Especially the flow of the language. I should start with a short introduction, objectives, and major conclusions.
- Your introduction needs more information about performance. I missing some general information regarding the performance of asphalt mixtures, such as https://doi.org/10.1016/j.conbuildmat.2017.07.164 and doi.org/10.1080/14680629.2021.1908408.
- Please state clearly your objectives.
- Please provide mix design data.
- You don`t have real conclusions. Don`t repeat your discussion. You did a lot of work so you should have good conclusions.
Round 2
Reviewer 3 Report
Thank you for addressing my comments.